# Cohort profile: maternal antecedents of adiposity and studying the transgenerational role of hyperglycaemia and insulin (MAASTHI)

Eunice Lobo,[1] Yamuna Ana,[1] R Deepa,[1] Prafulla Shriyan,[1] N D Sindhu,[1] Maithili Karthik,[1] Sanjay Kinra,[2] G V S Murthy,[3] Giridhara R Babu [4]

[1]Indian Institute of Public Health, Public Health Foundation, Bangalore, Karnataka, India
[2]Department of Non-Communicable Diseases Epidemiology, London School of Hygiene & Tropical Medicine, London, UK
[3]IIPH Hyderabad, Public Health Foundation, Hyderabad, Telangana, India
[4]Department of Population Medicine, College of Medicine, QU Health, Qatar University, Doha, Qatar

**Correspondence to**
Dr Giridhara R Babu;
epigiridhar@gmail.com

## ABSTRACT

**Purpose** The Maternal Antecedents of Adiposity and Studying the transgenerational role of Hyperglycaemia and Insulin cohort in Bengaluru, South India, aims to understand the transgenerational role of increased circulating glucose levels or hyperglycaemia and other nutrients and psychosocial environment, on the risk of childhood obesity, as an early marker of chronic diseases.

**Participants** Through this paper, we describe the baseline characteristics of the cohort participants and their children, along with plans and challenges. A total of 5694 pregnant women were screened, with 4862 (85.4%) eligible pregnant women recruited at baseline. We assessed anthropometry, Haemoglobin status, Oral Glucose Tolerance Test (OGTT), dietary practices, depressive symptoms using the Edinburgh Postnatal Depression Scale and social support in all women. Follow-up visits involved assessing anthropometry and the health profile of mothers and children.

**Findings to date** Among 4862 eligible participants recruited, 3260 (67%) underwent OGTT, while 2962 participants completed OGTT (90.9%). During the pregnancy, 9.7% of women were obese (>90th percentile of skinfold thickness), and 14.3% had gestational diabetesmellitus. Moreover, 6.2% and 16.8% of women had symptoms suggestive of depression during pregnancy and the immediate postnatal period, respectively. We found that 3.3% of children were small for gestational age, 10.8% were large for gestational age and 9.7% of children were obese at birth.

**Future plans** We have completed recruitment and baseline data collection in 2019, and are conducting annual follow-ups until age 4 of the participant's children. For delineating causal pathways of childhood obesity, blood aliquots are stored in the biorepository. The study will inform policy formulation and community awareness in the prevention and control of non-communicable diseases and health promotion.

## INTRODUCTION

According to the recent global disease burden study, type 2 diabetes mellitus (T2DM) is the fifth most common disease affecting Indians.[1] T2DM is projected to affect 70 million Indians by the year

---

## STRENGTHS AND LIMITATIONS OF THIS STUDY

⇒ Maternal Antecedents of Adiposity and Studying the transgenerational role of Hyperglycaemia and Insulin is a well-established cohort study documenting the life course trajectory in the genesis of non-communicable diseases (NCDs) for important exposure contrasts experienced by women very early on in their pregnancy and continues to document the outcomes across the process of pregnancy, childbirth, and early childhood, including assessing important milestones of the child.

⇒ This is the largest prospective birth cohort in the urban public sector on maternal and child health in South India, with a large sample size over a relatively long period of follow-up.

⇒ Blood aliquots are stored in the biorepository for future studies on childhood obesity.

⇒ Operational issues in terms of incomplete oral glucose tolerance test and lost to follow-up after delivery are the limitations of conducting a study in public health facilities.

⇒ Our cohort will inform policy formulation and community awareness in the prevention and control of NCDs and health promotion.

---

2025[2–7] at a relatively younger age compared with high-income countries. The increasing prevalence of gestational diabetes mellitus (GDM) directly impacts the T2DM burden.[6] Findings from the Hyperglycaemia and Adverse Pregnancy Outcome study initially confirmed the association between maternal glucose levels and neonatal adiposity.[8–11] These results conform to the theoretical frameworks of 'fuel-mediated teratogenesis', 'thrifty phenotype' and thrifty genotype' hypotheses,[2 8 11–15] which explain the transgenerational nature of obesity-hyperglycaemia. Even the Parthenon Birth cohort and others[16–18] have provided initial pieces of evidence regarding this association from low-income and middle-income countries

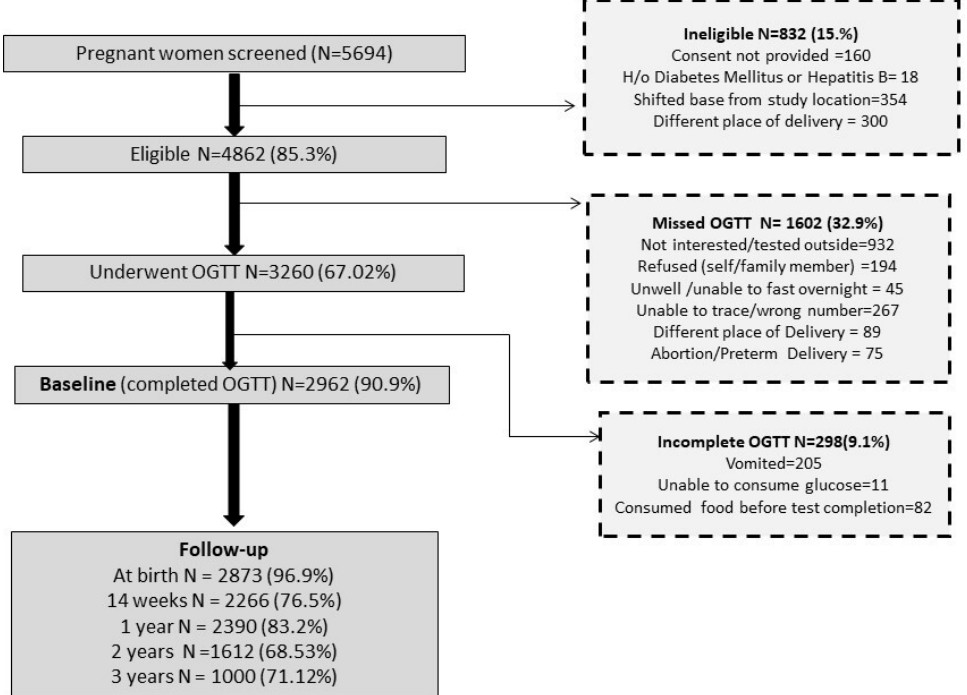

**Figure 1** Study recruitment and follow-ups in the MAASTHI cohort in India, 2016–2019. The flow chart shows the recruitment to follow-up of participants into the MAASTHI cohort and the reasons for drop-out. MAASTHI, Maternal Antecedents of Adiposity and Studying the transgenerational role of Hyperglycaemia and Insulin; OGTT, Oral Glucose Tolerance Test.

(LMICs). Thus, it becomes necessary to update the recent findings and, more importantly, capture the evidence from resource-constrained, high-burden countries such as India. Given the rapid demographical and epidemiological transition in India, there is a pressing need to capture the evidence on the obesity-hyperglycaemia burden.[19]

In LMICs, children who have low birth weight (LBW) and early undernutrition are known to have rebound adiposity during early adolescence.[20] Therefore, it is important to capture the life course trajectories of the underweight and obesity-led hyperglycaemia epidemics in LMICs. The risk factors of LBW may also be modifiable to prevent ensuing obesity epidemics in these countries. Among these risk factors, undernutrition in pregnancy affects nearly 28% of newborns having LBW in South Asia.[21 22] Evidence including high-income countries suggests a putative role in the psychosocial environment and weight of the newborn.[23 24] In addition, a recent systematic review also showed similar results for LMICs, with risk factors including poor social support, history of common mental disorders, etc.[25]

Individual and household stressors frequent exposure leads to poor mental health, including depression and anxiety.[26] The mother's poor mental health throughout pregnancy or after delivery affects mother–child interactions and makes it more difficult for child upbringing.[27] The effect of maternal mental health on child growth is likely to be more apparent in low socioeconomic situations due to the relative lack of resources and challenging contextual factors that are associated with poverty.

Aligned with the evidence from the high-income countries, we piloted the feasibility of conducting a multi-centric cohort study in the urban public health facilities of Bengaluru. Informed from the results of the pilot study,[28] the principal investigator (GRB) obtained funding to set up a birth cohort titled 'Maternal Antecedents of Adiposity and Studying the transgenerational role of Hyperglycaemia and Insulin' (MAASTHI). The primary objective of the cohort is to investigate the effect of glucose levels in pregnancy on skinfold thickness (SFT) (adiposity) in infancy as a marker of future obesity and diabetes in offspring, while the secondary objective is to assess the association between the psychosocial environment of mothers and adverse neonatal outcomes including adiposity. By capturing evidence from the mothers and children of low-income and middle-income status visiting health facilities in the public sector, MAASTHI will provide a contextual evidence base resulting in obesity-hyperglycaemia in India.[9 28]

## COHORT DESCRIPTION
### Study design, sample size, study site and participants
We started the recruitment in the prospective cohort study in April 2016 in the urban public health facilities of Bengaluru, a metropolitan city with a 13 million population in South India. The study protocol with the details of recruitment and methodology is published elsewhere.[29] With the assumption of a 5% incidence of childhood obesity at birth in India, a relative risk of 1.5 in the hyperglycaemic group, our estimated sample size with 80% power required to detect a difference at a 95% confidence level was

**Table 1** Assessments and data collected among the participants of MAASTHI birth cohort, Bengaluru 2016

| Time | Variable |
|---|---|
| Mother: 14–36 weeks | Sociodemographic details<br>Family medical history<br>Physical activity<br>Tobacco/alcohol use (self and spouse)<br>Dietary habits/24-hour recall<br>Obstetrics history<br>Assessment of depressive symptoms (using EPDS) and social support<br>Medications/supplements taken<br>Blood pressure<br>Anthropometry measurements<br>▶ Weight<br>▶ Height<br>▶ Biceps skinfold thickness<br>▶ Triceps skinfold thickness<br>▶ Subscapular skinfold thickness<br>▶ Head circumference<br>▶ Mid upper arm circumference (MUAC) |
| 24–36 weeks | OGTT (fasting plasma glucose, 2-hour postprandial plasma glucose) haemoglobin |
| At delivery | Delivery details<br>Mother's weight<br>Cause of death, if any<br>Random blood sugar for GDM women<br>Initiation of breastfeeding and feeding practice<br>Assessment of depressive symptoms (using EPDS) |
| Postdelivery—14 weeks, 1, 2, 3, 4 years | Health status<br>Medications/supplements taken<br>Body mass index (BMI)<br>Blood pressure<br>Assessment of depressive symptoms (using EPDS)<br>Social support<br>Anthropometry measurements<br>▶ Weight<br>▶ Waist circumference<br>▶ Hip circumference<br>▶ Biceps skinfold<br>▶ Triceps skinfold<br>▶ Subscapular skinfold |
| Child: At birth, 14th week, 1, 2, 3, 4 years | BMI<br>Anthropometry measurements<br>▶ Weight<br>▶ Length<br>▶ Crown-rump length (birth to 1 year)<br>▶ Head circumference<br>▶ Chest circumference<br>▶ Waist circumference<br>▶ Hip circumference<br>▶ MUAC<br>▶ Biceps skinfold thickness<br>▶ Triceps skinfold thickness<br>▶ Subscapular skinfold thickness<br>Infant data and neonatal problems<br>Morbidity/disease/neonatal intensive care unit/paediatric intensive care unit admissions<br>Immunisation records<br>Medications and supplement use<br>Diet<br>Milk or formula feeding<br>Introduction of complementary food and reason<br>Cessation of breast feeding and reason |
| 14 weeks, 1, 2, 3, 4 years | Trivandrum developmental screening chart |
| 2 years | Toddler feeding practice<br>Modified checklist for autism in toddlers<br>Physical activity |

Fasting plasma glucose equal to or more than 92 mg/dL or 2-hour postprandial plasma glucose equal to or more than 152 mg/dL(32) as GDM.
BMI, body mass index; EPDS, Edinburgh Postnatal Depression Scale; GDM, gestational diabetes mellitus; MAASTHI, Maternal Antecedents of Adiposity and Studying the transgenerational role of Hyperglycaemia and Insulin; OGTT, Oral Glucose Tolerance Test.

2936 participants. Considering lost to follow-up by about 60%, we had aimed to recruit 5000 pregnant women. We started recruitment in a 300-bedded public health facility (Jayanagar General Hospital) in Bengaluru corporation. The cohort is established in 14 public health facilities in Bengaluru, Karnataka viz. Jayanagar General Hospital, Bagaluru Referral Hospital, Srirampura Referral Hospital, Anjanappa Garden UPHC, Banashankari Referral Hospital, Bapuji Nagar UPHC, D J Halli UPHC, KC General Hospital, Magadi road UPHC, Pantharapalaya UPHC, Robertsonpet

**Table 2** Baseline characteristics of eligible pregnant mothers and children in the MAASTHI birth cohort, Bengaluru, 2016–2021

| Maternal characteristics | |
|---|---|
| Sociodemographic characteristics | n (%) |
| Age (years) (N=4862) | |
| Mean age±SD | 24.25±4.06 |
| 18–25 | 3211 (66.0) |
| 26–35 | 1598 (32.9) |
| 36–45 | 53 (1.1) |
| Religion (N=4862) | |
| Hindu | 2454 (50.5) |
| Muslim | 2142 (44.1) |
| Others | 266 (5.5) |
| Consanguineous marriage (N=4788) | 1157 (24.2) |
| Participant's education (N=4788) | |
| Illiterate | 169 (3.5) |
| Primary school | 284 (5.9) |
| Middle school | 609 (12.7) |
| High school | 2091 (43.7) |
| PUC or diploma | 1072 (22.4) |
| Graduate and above | 563 (11.8) |
| Husband's education (N=4788) | |
| Illiterate | 426 (8.9) |
| Primary school | 461 (9.6) |
| Middle school | 724 (15.1) |
| High school | 1956 (40.9) |
| PUC or diploma | 793 (16.6) |
| Graduate and above | 422 (8.8) |
| Do not know | 6 (0.1) |
| Participant's occupation (N=4788) | |
| Homemaker | 4446 (92.9) |
| Unskilled worker | 168 (3.5) |
| Semiskilled worker | 76 (1.6) |
| Skilled worker | 85 (1.8) |
| Professional | 13 (0.3) |
| Husband's occupation (N=4788) | |
| Unemployed | 17 (0.4) |
| Unskilled worker | 2237 (46.7) |
| Semiskilled worker | 1535 (32.1) |
| Skilled worker | 860 (18.0) |
| Professional | 139 (2.9) |
| Socioeconomic status (N=4788) | |
| Lower class | 2614 (54.6) |
| Middle class | 2145 (44.8) |
| Upper class | 29 (0.6) |
| Chronic conditions with 95% CI of prevalence estimates | |
| Hypertension (N=3719) | |
| Normal | 3429 (92.2) (95% CI: 0.91 to 0.93) |

Continued

**Table 2** Continued

| Maternal characteristics | |
|---|---|
| Pre hypertension | 265 (7.1) (95% CI: 0.69 to 0.72) |
| Stage 1 hypertension | 15 (0.4) (95% CI: 0.0020 to 0.0060) |
| GDM during current pregnancy (N=2962) | 424 (14.3) (95% CI: 0.13 to 0.16) |
| Anaemia (N=3097) | 1377 (44.5) (95% CI: 0.42 to 0.46) |
| Obstetrical characteristics (N=4778) | |
| Gravida | |
| Primigravida | 1920 (40.2) |
| Multigravida | 2858 (59.8) |
| Parity | |
| Nulliparous | 2178 (45.6) |
| Multiparous | 2600 (54.4) |
| Mean Gestational age at delivery (week) (N=3324) | 38.70±3.02 |
| Depressive symptoms and social support | |
| EPDS score at pregnancy (>13 cut-off) (N=4635) | 289 (6.2) |
| Social support score above 24 (N=4643) | 1520 (32.7) |
| EPDS score at delivery (>13 cut-off) (N=2801) | 471 (16.8) |
| Health status of mothers at delivery (N=3324) | |
| Healthy | 3054 (92.0) |
| Morbidity/hospitalisation* Death | 259 (7.8) |
| | 7 (0.2) |
| Child characteristics n (%) | |
| Health status at birth (N=3310) | |
| Healthy | 3188 (96.3) |
| Morbidity/hospitalisation† | 49 (1.5) |
| Death | 73 (2.2) |
| Preterm deliveries (N=4862) | 430 (8.8) |
| Birth weight (N=4862) | |
| Underweight <2500 g | 384 (7.9) |
| Normal 2500 g to 3500 g | 1625 (33.4) |
| Overweight >3500 g | 2853 (58.7) |
| Birth weight according to Gestational Age (N=4862) | |
| Small for Gestational Age (SGA) | 159 (3.3) |
| Mean±SD wt for SGA (kg) | 2.13±0.26 |
| Large for gestational age (LGA) | 523 (10.8) |
| Mean±SD wt for LGA (kg) | 4.10±0.72 |
| Delivery type (N=3324) | |
| Vaginal delivery | 1868 (56.2) |
| Caesarean section | 1456 (43.8) |
| Sex (N=3309) | |
| Male | 1717 (51.9) |
| Female | 1592 (48.1) |
| Baby cried soon after delivery (N=3310) | 3030 (91.5) |

Continued

**Table 2** Continued

| Maternal characteristics | |
|---|---|
| Colostrum fed to the baby (N=2915) | 2562 (87.9) |
| Breast milk fed as first food to the baby (N=2915) | 1932 (66.3) |
| Initiation of breast feeding after delivery (N=2890) | |
| Within an hour | 1227 (42.4) |
| Between 1 and 24 hours | 1538 (53.2) |
| After 24 hours | 125 (4.4) |

GDM: Fasting blood sugar equal to or more than 92 mg/dL or 2-hour postprandial blood sugar equal to or more than 152 mg/dL. Hypertension: normal: SBP <120 mm Hg and DBP <80 mm Hg; prehypertension: SBP 120–139 mm Hg or DBP 80–89 mm Hg; stage 1 hypertension: SBP 140–159 mm Hg or DBP 90–99 mm Hg.
*Morbidity/hospitalisation (mother): postpartum haemorrhage, anaemia, infections, hypertensive disorder, diabetes mellitus, respiratory illness, thyroid disease.
†Morbidity/hospitalisation (newborn): fever, admission to NICU due to jaundice, low birth weight, IUGR, allergy, asphyxia, respiratory distress syndrome, Gastrointestinal abnormalities, congenital anomaly, macrocephaly, birth trauma, meconium aspiration, premature baby, infections, convulsions, pneumonia, hypoglycaemia.
DBP, diastolic blood pressure; EPDS, Edinburgh Postnatal Depression Scale; GDM, gestational diabetes mellitus; IUGR, Intrauterine growth restriction; MAASTHI, Maternal Antecedents of Adiposity and Studying the transgenerational role of Hyperglycaemia and Insulin; NICU, neonatal intensive care unit; PUC, Pre-university college; SBP, systolic blood pressure.

UPHC, Siddaiah Referral Hospital, Sirsi Circle UPHC and Tavrekere UPHC.

All pregnant women between 14 and 36 weeks of gestation from the ages of 18–45 years visiting the public health facilities were approached to participate in the study. After obtaining written informed consent, we recruited the women only if they planned to deliver in the study health facility and were available for future follow-ups in Bengaluru. Exclusion criteria included a history of major co-existing diseases as HIV/ADIS, hepatitis B, diabetes and the inability to complete the oral glucose tolerance test (OGTT) within 24–36 weeks of gestation.

### Patient and public involvement
No patient involvement.

### Data collection
#### Recruitment and follow-up
We approached 5694 pregnant women and collected the baseline data from 4862 eligible pregnant women and 2873 women who delivered during the period of April 2016–June 2019. A brief overview of study recruitment, retention and at-birth follow-up is presented in figure 1.

After obtaining written informed consent, eligible pregnant women were recruited into the study when they completed 14 weeks of gestational age, and face-to-face interviews were done by trained research assistants ensuring privacy and confidentiality.

### What has been measured?
We have collected sociodemographic details, 24-hour dietary recall, dietary habits, use of tobacco and alcohol (own and spouse) and physical activity in the women. In addition to obtaining a family history of diabetes and other cardiovascular diseases, we obtained an obstetric history from all the pregnant women (table 1). In addition, we have assessed the psychosocial environment using validated versions of the Edinburgh Postnatal Depression Scale (EPDS) and social support scale.[30] Between 24 and 36 weeks, all the women were invited to undergo OGTT. GDM was classified using the WHO diagnostic criteria.[31] We also measured blood pressure (BP), assessed body mass index (BMI) and measured haemoglobin in all the women (table 1).

The follow-up visits began soon after the delivery. We recorded the health status, morbidities or any other illness of the baby, feeding and mother's follow-up assessments. The measurements of SFT, circumferences and weight, length in infants were taken at birth and 14 weeks, corresponding to the child's immunisation visits to the public health facility. Further follow-up visits of the children include anthropometric assessment annually at ages 1, 2, 3 and 4; along with other assessments mentioned in table 1.

We use the support from frontline health workers in the community to remind and help track the unavailable participants for follow-up visits. In addition, the research staff regularly call the participants for reminders regarding the follow-up schedule, and if required, also do house visits (after prior permission). These additional efforts are done for participants who cannot visit the public health facility during the defined period. To make it easier for participants, we also schedule follow-up visits in the primary health centres and Anganwadi centres near their residence. We have also examined the role of an interactive voice-based response system (IVRS) and an information workshop in improving the follow-up rates in these women.[32]

Through IVRS, women received a 2–3 min call to inform them about OGTT test dates, collect their laboratory test reports, and remind them of their follow-up visits. The IVRS also provided health information on GDM, breast feeding and immunisation.

A mother and baby Affairs workshop or antenatal workshop and counselling for parents were conducted. This included a lunch for the participant and her family (husband and children, if any). Generally, this is called 'Seemantha' in Kannada, similar to a baby shower in other parts of the world.

The workshop included a brief talk by health professionals on the importance of GDM screening and management and other antenatal and postnatal topics.

### Storage of blood samples for future analysis
At 24–36 weeks, during the collection of blood for OGTT, we collect 11 mL venous blood samples in the fasting state and 2 mL for postprandial blood glucose analysis.

**Table 3** Anthropometry of the mother at different follow-up periods of the MAASTHI birth cohort, Bengaluru, 2016–2021

| Anthropometric measurement | | During pregnancy (N=3734) | | 14th week follow-up (N=2146) | | 1-year follow-up (N=1255) | | 2-year follow-up (N=801) | | 3-year follow-up (N=393) | |
|---|---|---|---|---|---|---|---|---|---|---|---|
| | | n (%) | Mean (SD) | n (%) | Mean (SD) | n (%) | Mean (SD) | n (%) | Mean (SD) | n (%) | Mean (SD) |
| Weight | <10th percentile | 367 (9.8) | 41.88 (2.54) | 214 (10.0) | 40.35 (2.73) | 124 (9.9) | 38.96 (2.21) | 80 (10.0) | 40.25 (2.91) | 39 (9.9) | 40.41 (2.04) |
| | 10–90th percentile | 2996 (80.2) | 58.09 (7.60) | 1720 (80.1) | 56.90 (7.76) | 1006 (80.2) | 56.54 (8.00) | 642 (80.1) | 59.25 (8.18) | 315 (80.2) | 59.27 (8.31) |
| | >90th percentile | 371 (9.9) | 82.74 (7.07) | 212 (9.9) | 81.27 (6.56) | 124 (9.9) | 81.79 (7.02) | 79 (9.9) | 85.86 (8.51) | 39 (9.9) | 83.84 (7.07) |
| Waist circumference | <10th percentile | – | – | 194 (9.6) | 61.51 (3.71) | 124 (9.9) | 60.18 (2.97) | 80 (10.0) | 61.04 (2.71) | 38 (9.7) | 60.27 (3.65) |
| | 10–90th percentile | – | – | 1635 (80.8) | 77.96 (7.51) | 1006 (80.2) | 77.43 (7.61) | 641 (80.0) | 79.87 (8.41) | 319 (81.2) | 79.01 (8.06) |
| | >90th percentile | – | – | 195 (9.6) | 99.46 (6.23) | 125 (10) | 98.99 (6.15) | 80 (10.0) | 103.59 (6.00) | 36 (9.2) | 98.82 (4.07) |
| MUAC | <10th percentile | 358 (9.6) | 20.01 (1.15) | 190 (8.8) | 20.57 (1.02) | 124 (9.9) | 20.39 (1.19) | 74 (9.2) | 20.59 (1.44) | 39 (9.9) | 20.44 (1.41) |
| | 10–90th percentile | 3002 (80.4) | 25.78 (2.55) | 1744 (81.2) | 26.42 (2.52) | 1012 (80.6) | 26.74 (2.62) | 649 (81.0) | 27.65 (2.68) | 315 (80.2) | 27.30 (2.65) |
| | >90th percentile | 372 (10.0) | 33.54 (2.11) | 213 (9.9) | 34.03 (2.08) | 119 (9.5) | 43.43 (2.05) | 78 (9.7) | 35.30 (2.05) | 39 (9.9) | 34.44 (1.66) |
| Total SFT | <10th percentile | 361 (9.7) | 25.21 (2.91) | 213 (9.9) | 24.72 (3.51) | 124 (9.9) | 22.44 (3.38) | 79 (9.9) | 25.37 (4.19) | 39 (9.9) | 24.19 (4.12) |
| | 10–90th percentile | 3008 (80.6) | 46.54 (10.05) | 1723 (80.3) | 46.84 (9.59) | 1007 (80.2) | 45.93 (10.17) | 641 (80.1) | 50.69 (10.20) | 315 (80.2) | 49.79 (10.17) |
| | >90th percentile | 363 (9.7) | 76.60 (8.31) | 210 (9.8) | 73.83 (6.75) | 125 (10.0) | 73.92 (6.04) | 80 (10.0) | 79.55 (8.15) | 39 (9.9) | 75.46 (7.00) |
| Hip Circumference | <10th percentile | – | – | 199 (9.8) | 79.39 (4.85) | 121 (9.6) | 77.52 (4.00) | 80 (10.0) | 78.71 (5.93) | 39 (9.9) | 80.33 (3.29) |
| | 10–90th percentile | – | – | 1624 (80.2) | 94.99 (6.71) | 1013 (80.7) | 94.75 (7.06) | 642 (80.1) | 97.18 (7.16) | 315 (80.2) | 96.69 (6.89) |
| | >90th percentile | – | – | 201 (9.9) | 115.35 (5.47) | 121 (9.6) | 115.47 (5.71) | 79 (9.9) | 118.56 (5.75) | 39 (9.9) | 115.63 (3.29) |
| Height | <10th percentile | 368 (9.9) | 144.15 (2.27) | – | – | – | – | – | – | – | – |
| | 10–90th percentile | 2993 (80.2) | 153.81 (3.80) | – | – | – | – | – | – | – | – |
| | >90th percentile | 373 (10.0) | 164.16 (2.77) | – | – | – | – | – | – | – | – |
| Head circumference | <10th percentile | 329 (8.8) | 49.46 (1.09) | – | – | – | – | – | – | – | – |
| | 10–90th percentile | 3080 (82.5) | 52.69 (1.15) | – | – | – | – | – | – | – | – |
| | >90th percentile | 324 (8.7) | 56.08 (1.00) | – | – | – | – | – | – | – | – |

MAASTHI, Maternal Antecedents of Adiposity and Studying the transgenerational role of Hyperglycaemia and Insulin; MUAC, mid-upper arm chest; SFT, skinfold thickness.

**Table 4** Anthropometry of the child at different follow-up periods of the MAASTHI birth cohort, Bengaluru, 2016–2021

| Anthropometric measurement | | At birth | | 14-week follow-up | | 1-year follow-up | | 2-year follow-up | | 3-year follow-up | |
|---|---|---|---|---|---|---|---|---|---|---|---|
| | | n (%) | Mean (SD) | n (%) | Mean (SD) | n (%) | Mean (SD) | n (%) | Mean (SD) | n (%) | Mean (SD) |
| Weight | <10th percentile | 243 (9.8) | 2.14 (0.21) | 182 (8.5) | 4.31 (0.39) | 122 (9.7) | 6.84 (0.38) | 74 (9.0) | 8.56 (0.62) | 41 (10.0) | 9.80 (0.64) |
| | 10–90th percentile | 2003 (80.5) | 2.98 (0.41) | 1761 (82.4) | 6.04 (0.72) | 1010 (80.5) | 8.74 (0.82) | 670 (81.5) | 10.73 (0.90) | 333 (81.2) | 12.30 (0.95) |
| | >90th percentile | 242 (9.7) | 4.78 (0.74) | 195 (9.1) | 8.37 (0.73) | 122 (9.7) | 11.43 (0.85) | 78 (9.5) | 13.41 (0.77) | 36 (8.8) | 14.74 (0.28) |
| CRL | <10th percentile | 233 (9.4) | 27.37 (2.01) | 213 (10.0) | 33.21 (1.89) | 114 (9.9) | 37.28 (2.48) | – | – | – | – |
| | 10–90th percentile | 2020 (81.5) | 32.41 (2.01) | 1719 (80.6) | 40.52 (2.40) | 926 (80.2) | 45.31 (2.36) | – | – | – | – |
| | >90th percentile | 226 (9.1) | 39.29 (2.21) | 202 (9.5) | 47.66 (2.79) | 114 (9.9) | 53.44 (4.00) | – | – | – | – |
| Length | <10th percentile | 248 (10.0) | 43.69 (1.72) | 206 (9.6) | 55.26 (2.22) | 125 (10.0) | 65.76 (6.28) | 79 (9.6) | 74.64 (3.09) | 41 (10.0) | 83.16 (3.42) |
| | 10–90th percentile | 2009 (80.8) | 49.38 (2.28) | 1720 (80.5) | 62.66 (2.74) | 1003 (80.2) | 73.79 (2.65) | 664 (80.8) | 83.17 (2.92) | 329 (80.2) | 90.78 (2.77) |
| | >90th percentile | 230 (9.2) | 58.06 (2.81) | 211 (9.9) | 70.99 (2.24) | 123 (9.8) | 81.68 (2.78) | 79 (9.6) | 91.61 (2.23) | 40 (9.8) | 97.89 (1.65) |
| Head circumference | <10th percentile | 242 (9.7) | 30.30 (1.07) | 199 (9.3) | 36.30 (2.05) | 102 (8.1) | 40.39 (1.79) | 81 (9.9) | 42.49 (1.45) | 39 (9.5) | 43.96 (0.87) |
| | 10–90th percentile | 2001 (80.4) | 33.66 (1.33) | 1740 (81.4) | 40.07 (1.33) | 1033 (82.4) | 44.23 (1.16) | 666 (81.3) | 46.16 (1.25) | 332 (81.0) | 47.10 (1.09) |
| | >90th percentile | 245 (9.8) | 38.10 (1.25) | 199 (9.3) | 43.97 (0.96) | 119 (9.5) | 47.41 (1.33) | 72 (8.8) | 50.81 (4.86) | 39 (9.5) | 50.27 (1.48) |
| Chest circumference | <10th percentile | 246 (9.9) | 28.10 (1.19) | 198 (9.3) | 35.67 (1.02) | 125 (10.0) | 39.73 (1.89) | 69 (8.4) | 42.54 (1.41) | 38 (9.3) | 44.57 (0.87) |
| | 10–90 percentile | 2014 (80.9) | 32.24 (1.73) | 1740 (81.4) | 40.12 (1.71) | 1017 (81.1) | 44.35 (1.58) | 673 (81.8) | 46.78 (1.58) | 333 (81.2) | 48.56 (1.69) |
| | >90th percentile | 228 (9.2) | 38.38 (1.83) | 200 (9.4) | 45.08 (1.27) | 112 (8.9) | 49.15 (1.48) | 81 (9.8) | 56.49 (1.64) | 39 (9.5) | 53.0 (1.04) |
| Waist circumference | <10th percentile | 235 (9.4) | 24.81 (1.17) | 182 (8.5) | 33.14 (1.46) | 2 (0.2) | 29.30 (1.69) | 77 (9.3) | 40.17 (1.64) | 41 (10.0) | 41.93 (1.20) |
| | 10–90 percentile | 2008 (80.7) | 30.66 (2.47) | 1747 (81.7) | 39.28 (2.29) | 1132 (90.3) | 41.95 (2.86) | 667 (80.9) | 45.57 (1.98) | 329 (80.2) | 46.94 (2.09) |
| | >90th percentile | 245 (9.8) | 38.24 (1.87) | 209 (9.8) | 45.37 (1.58) | 119 (9.5) | 49.02 (1.83) | 80 (9.7) | 51.39 (1.96) | 40 (9.8) | 53.27 (1.40) |
| Hip circumference | <10th percentile | 211 (8.5) | 23.67 (0.93) | 211 (9.9) | 32.05 (1.49) | 4 (0.3) | 30.75 (2.12) | 79 (9.6) | 39.71 (1.47) | 39 (9.5) | 41.70 (4.69) |
| | 10–90 percentile | 2037 (81.9) | 28.91 (2.44) | 1717 (80.3) | 39.05 (2.85) | 1125 (89.7) | 42.17 (3.06) | 669 (81.2) | 46.23 (2.51) | 332 (81.0) | 48.71 (2.50) |
| | >90th percentile | 240 (9.6) | 37.32 (2.68) | 210 (9.8) | 46.42 (1.60) | 125 (10.0) | 50.0 (1.97) | 76 (9.2) | 53.47 (1.87) | 39 (9.5) | 55.21 (1.31) |
| MUAC | <10th percentile | 191 (7.7) | 7.86 (0.58) | 206 (9.6) | 10.58 (0.76) | 124 (9.9) | 11.25 (0.96) | 78 (9.5) | 11.65 (0.82) | 39 (9.5) | 12.28 (1.47) |
| | 10–90 percentile | 2074 (83.4) | 9.81 (0.76) | 1748 (81.8) | 12.88 (0.82) | 1015 (80.9) | 13.77 (0.84) | 665 (80.8) | 14.15 (0.80) | 333 (81.2) | 14.57 (0.73) |
| | >90th percentile | 223 (9.0) | 12.48 (0.77) | 183 (8.6) | 15.31 (0.70) | 115 (9.2) | 16.25 (1.52) | 80 (9.7) | 16.72 (1.03) | 38 (9.3) | 17.76 (5.78) |
| Total SFT | <10th percentile | 223 (9.0) | 9.27 (0.78) | 196 (9.2) | 14.25 (1.20) | 119 (9.5) | 14.36 (0.98) | 78 (9.5) | 14.94 (1.20) | 40 (9.8) | 15.53 (1.06) |
| | 10–90 percentile | 2005 (81.0) | 13.76 (2.14) | 1723 (81.2) | 20.80 (2.61) | 1006 (80.5) | 19.98 (2.47) | 670 (81.4) | 21.58 (3.05) | 332 (81.0) | 22.87 (3.41) |
| | >90th percentile | 246 (9.9) | 22.37 (3.25) | 204 (9.6) | 29.45 (3.31) | 124 (9.9) | 28.02 (2.74) | 75 (9.1) | 32.46 (3.74) | 38 (9.3) | 32.60 (3.28) |

MAASTHI, Maternal Antecedents of Adiposity and Studying the transgenerational role of Hyperglycaemia and Insulin; MUAC, mid-upper arm chest; SFT, skinfold thickness.

The blood is collected in three vacutainers; plain (6 mL for storage), EDTA (3 mL for haemoglobin analysis), and sodium fluoride vacutainers (2 mL each for glucose analysis in fasting and postprandial), respectively. Blood samples are centrifuged and transferred to cryovials within an hour of collecting in cool boxes to a single central laboratory for assays with external quality assurance mechanisms.

## Measurements

We perform anthropometric measurements of mothers at the baseline and during subsequent follow-up visits. The weight was measured using Tanita weighing scale, and height using the SECA 213 portable stadiometer. The mother's BP is measured using the automated digital device (Omron Digital BP measuring device). The anthropometric measurements of the baby are recorded within 72 hours of delivery. Newborn anthropometry was performed using SECA 354 Weighing Scale and SECA 417 infantometer. We measure the head, mid-upper arm, chest and waist circumferences in both the mother and child using Chasmors body circumference tape. The sum of SFT (biceps, triceps and subscapular) (SFT) is measured on the left side of the body using the Holtain Callipers (Holtain, UK).

## Quality control

Research assistants were trained and certified by St. Johns Research Institute, Bengaluru, in anthropometric measurements. Strict protocols are followed to maintain accuracy, with the addition of interobserver and intraobserver reliability of measurements assessed at the outset followed by annual certification by the same institute. Trained phlebotomists collect venous blood for laboratory investigations. Biochemical assays are conducted at a central nationally accredited laboratory with internal and external quality checks. Calibration of all the equipments is done every month using prescribed guidelines, with a calibration log maintained by the research staff and supervised by the principal investigator of the study.

## Data management and analysis

Data are entered on android tablets regularly using the application specifically designed for the study. The details of the development and use are published elsewhere.[33] The primary data collected is exported through MS Excel 2010 and checked periodically by senior team members for data entry errors and missing information. For this paper, the exported data were then cleaned, and explorative data analysis was done to understand and summarise the variables involved. The descriptive analysis was performed using SPSS V.23. The modified EPDS scores cut-off for defining symptoms suggestive of depression was 13 based on a previous study in Karnataka,[34] and social support scores were categorised based on the cut-off score of 24 and above as good(≥24) and poor(<24).[30] Adiposity (mother and child) and gestational age

(child) weight were classified based on percentiles— using 90th percentile as the cut-off value. Adiposity was considered based on the total sum of SFT (using biceps, triceps and subscapular thickness). At the same time, weight for gestational age was classified considering the weight of children at each gestational age, adjusted for parity and gender.[35] The birth weight of children was categorised based on the WHO standards, and preterm births were defined as gestational age less than 37 weeks.[36] Anaemia in pregnant women was categorised on the WHO criteria based on haemoglobin concentration: Anaemic when <110 g/L at sea level, and no anaemia when above 100 g/L.[37] Hypertension among participants was categorised into: normal (<120/80 mm Hg), prehypertension (120–139/80–89 mm Hg), stage 1 hypertension (140–159/90–99 mm Hg), and stage 2 hypertension (≥160/100 mm Hg).[38] BMI of the participants was categorised based on the WHO Asian criteria: underweight (<18.5), normal (18.5–22.9), overweight (23.0–24.9) and obese (>25.0).[39] Socioeconomic status was categorised using the modified Kuppuswamy socioeconomic status scale 2017 that classifies socioeconomic status as: lower, middle and upper classes.[40]

## Findings to date and discussion

Among 5694 pregnant women, 85.3% (n=4862) were included in the study based on the eligibility criteria and willingness to participate. A total of 67% of women underwent OGTT (n=3260) as per the schedule between 24 and 36 weeks, with a completion rate of 90.9% (n=2962) (figure 1). At the baseline, the mean age of the pregnant women was 24.25±4.06 years, with 66% between the age group of 18–25 years. Most pregnant women had high school education (66%; n=437). More than 90% of women were homemakers (90.0%; n=4446), while most of their spouses were unskilled workers (46.7%; n=2237%). Using the modified Kuppuswamy scale, most participants were categorised as lower socioeconomic groups (54.6%; n=2614) (table 2).

We found the prevalence of GDM in these women was 14.3% with 95% CI: 0.13 to 0.16 (n=424/2962). Among the recruited participants, 265 were prehypertensive (7.1%; 95% CI: 0.69% to 0.72%) and 15 were hypertensive (0.4%; 95% CI: 0.0020% to 0.0060%), while 44.5% of the women were anaemic (n=1377; 95% CI: 0.42 to 0.46). Using the cut-off value of 13 for EPDS, we found that 6.2% of women (n=289) had depressive symptoms indicative of depression during pregnancy and symptoms of postnatal depression were present in 16.8% of women (n=471). The mean gestational age at delivery was 38.7±3.02 weeks. More than half of the newborns were males (51.9%; n=1717), 43.8% of the infants were delivered by caesarean section (n=1456), and 91.2% infants were born full term (n=4432) (table 2).

Table 3 summarises the anthropometric measurements of the participants (mothers) and their offspring during each phase of the study period respectively. Categorisation

is based on tertiles –10th, 10– 90th and >90th percentiles. Each table includes mean and SD values for the measurements described in the analysis, and the frequency and proportion of mother and child in each tertile. We found that among the mothers above the 90th percentile for weight, 9.9% were obese at delivery and subsequent follow-ups. They weighed an average of 82.74 kg as compared with 83.84 kg at the 3 years follow-up. During the period from delivery to the 3-year follow-up, the average SFT in women with obesity (>90th percentile of SFT) changed from 76.6 mm (9.7%) to 75.46 mm (9.9%) in comparison to women with a normal range of SFT (10–90th percentile) in whom it changed from 46.54 mm (80.6%) to 49.79 mm (80.2%) (table 3).

Anthropometric measurements of children showed that the sum of SFT ranged from 22.37 mm to 32.6 mm from birth to the 3-year follow-up for children above the 90th percentile compared with 13.76 mm to 22.87 mm, respectively, in children between 10 and 90th percentile. Birth of a high proportion of obese children (weight above 90th percentile), that is, 9.7%, was noted in our study (table 4). We also noted that 7.9% of children born were LBW (2500 g or less) and 8.8% (n=430) were born preterm. A high proportion of 10.8% (n=523) of children born were large for gestational age (LGA) in our cohort, and 3.3% were small for gestational age (SGA) (n=159). The mean weight of children who were SGA at birth was 2.13±0.26 kg and was 4.10±0.72 kg for children with LGA.

### Strengths and limitations

As there is no unanimity regarding appropriate guidelines for diagnosing GDM in the Indian population, our findings can provide specific evidence regarding exact cut-off points to be used for the diagnosis. The high prevalence of GDM may provide evidence for a policy decision to scale up screening and management of GDM in all the public health facilities. Above this it may be associated with adverse maternal and childhood outcomes. Also, the huge burden of depressive symptoms among mothers, draws attention to the importance of mental health screening as part of routine antenatal care.

We found a high number of children born LGA. In addition, the prevalence of obesity at birth and subsequent follow-ups was also high (above 9%). If unchanged modifiable factors of obesity such as diet, exercise and GDM, will continue the vicious cycle of afflicting subsequent generations.[41] The results from our cohort will continue to inform and shape policies regarding screening, appropriate care and management for persons affected with GDM, perinatal depression and childhood obesity.

MAASTHI is a well-established cohort study documenting the life course trajectory in the genesis of non-communicable diseases (NCDs). We are documenting important exposure contrasts experienced by women early in their pregnancy and continue to document the outcomes across the process of pregnancy, childbirth

and early childhood, including assessing important milestones of the child. In India, public health facilities are the preferred choice for many women[42] from the lower social strata. Our findings revealed an unusually high burden of GDM and maternal obesity similar to earlier studies[43–46] for women between the age group of 18–45 years. Our cohort study will inform policies in achieving further reductions in maternal and fetal mortality and morbidity. For example, achieving the desired level of glycaemic control during pregnancy can reduce several maternal and fetal adverse effects.[47] Early screening and timely management of GDM may prevent some poor outcomes for mother and child.

The strength of MAASTHI is that data pertain to the pregnant women and children belonging to lower-income and middle-income socioeconomic status-seeking healthcare from the public health facilities with a strong representation of minority groups and vulnerable sections of the society. Also, this is probably the only pregnancy cohort recruited from public health facilities in India. Although the acronym suggests the transgenerational role of glucose and insulin, the study was conceptualised to include important maternal determinants of underweight and childhood obesity. The comprehensive exposure assessments in MAASTHI include the role of nutritional, psychosocial, and, recently more relevant—air pollution throughout pregnancy and different time points of the life course.[48] Attention to obtaining high-quality data by maintaining stringent quality checks for reproducible measurements and minimal errors is another major strength of this study. Moreover, blood aliquots are stored in the biorepository for future studies on childhood obesity.

Operational issues in terms of incomplete OGTT and lost to follow-up after delivery are the limitations of conducting a study in public health facilities. Without a registry or defined population for healthcare services, this limitation will continue to affect implementing cohort studies in public health facilities. Periods of prolonged fasting (previous night) coupled with the issue of gastritis/morning sickness in pregnant women are other factors responsible for incomplete OGTT. As with most research during COVID-19, our cohort was also subject to data collection issues. Our research staff had to resolve the inability of physical contact for face-to-face interviews through telephonic interviews. However, no physical meetings meant the complete stop to anthropometric measurements for mother–child dyads.

By setting up and following up this cohort, we can inform the design and implement interventions focusing on health promotion and disease prevention, address undernutrition and overweight epidemics, and thereby contribute to reducing the NCD disease profile in the country.

**Acknowledgements** We sincerely thank the Department of Health and Family Welfare, the Government of Karnataka (DoHFW, GoK) and the Bruhat Bengaluru

Mahanagara Palike (BBMP) for permitting us to conduct the study and providing constant support. We thank hospitals under DoHFW, GoK, Superintendents, Medical Officers, Doctors and all the department support staff for supporting an ongoing study. We thank Dr Sumathi Swaminathan and her team at St John's Research Institute for training the Research Assistants in anthropometry measurement. Our sincere thanks to the Welcome Trust, DBT India Alliance, for the support with funding and guidance. Our sincere thanks to Dr Suresh Shapeti and Mr TS Ramesh for facilitating the administrative support and coordination. We thank all the research team members of MAASTHI for their support in carrying out research activities in the field. We thank all participants for their effort to enroll and continuous participation in the ongoing cohort.

**Collaborators** The MAASTHI prospective cohort will continue to collect data on various exposures and maternal and child health related outcomes. For more information, refer to the website: maasthi.com. MAASTHI encourages proposals from outside investigators: Proposals involving the use of existing data or the collection of new data. Researchers interested in collaboration are invited to submit requests for MAASTHI clinical data through a research proposal to giridhar@phfi.org

**Contributors** GRB was involved in the conception, design, and funding acquisition. YA, RD, PS, MK and SN performed the experiments. EL and GRB were involved in drafting the paper and revising the draft. SK, GVSM were involved in the critical analysis of the paper. All authors have approved the final version of the article submitted. GRB is responsible for the overall content as the guarantor.

**Funding** This work was supported by the Wellcome Trust/DBT India Alliance Senior Fellowship (Grant No. IA/CPHS/20/1/505278] awarded to GRB.

**Disclaimer** The funding agency had no role in the design and conduct of the study, collection, management, analysis, and interpretation of the data, preparation, review, or approval of the manuscript, or decision to submit the manuscript for publication.

**Competing interests** None declared.

**Patient and public involvement** Patients and/or the public were not involved in the design, or conduct, or reporting, or dissemination plans of this research.

**Patient consent for publication** Not applicable.

**Ethics approval** This study involves human participants and was approved by Institutional ethics committee (IEC) of the Indian Institute of Public Health—Bengaluru Approval number IIPHHB/TRCIEC/091/2015. Participants gave informed consent to participate in the study before taking part.

**Provenance and peer review** Not commissioned; externally peer reviewed.

**Data availability statement** Data are available on reasonable request. To discuss our data sharing policy, please contact GRB at giridhar@phfi.org.

**ORCID iD**
Giridhara R Babu http://orcid.org/0000-0003-4370-8933

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
