## [Reviewer comments · BMJ Open]

ARTICLE DETAILS

TITLE (PROVISIONAL)	Cohort Profile: Maternal antecedents of adiposity and studying the transgenerational role of hyperglycemia and insulin (MAASTHI)
AUTHORS	Lobo, Eunice; Ana, Yamuna; Deepa, R; Shriyan, Prafulla; Sindhu, ND; Karthik, Maithili; Kinra, Sanjay; Murthy, GVS; Babu, Giridhara R

VERSION 1 – REVIEW

REVIEWER	Francis, Ellen University of Colorado - Anschutz Medical Campus
REVIEW RETURNED	01-Aug-2022

GENERAL COMMENTS	This cohort paper describes the measurement of exposures and outcomes in a longitudinal birth cohort study among a difficult to reach population. The undertaking of this study is commended, and the care taken to report the measurements, prevalence of exposures and outcomes, and implications is appreciated. There are a few minor revisions that I think will improve the MS. Strengths and limitations: Overall this section could be revised to be more streamlined. There are many redundant sentences. The vulnerability of the population and the implications of the study are repeated in almost every paragraph. This entire section could be substantially shortened. Introduction: Please provide a brief background into the interest in the secondary aims (e.g., depression). Tables: It may be worthwhile to provide a footnote with abbreviations and definitions. For example, what does "diseased" include, what criteria was used for GDM diagnosis? Why are there CIs for hypertension, GDM and anemia? Line 254 N is reported as a %. Line 271 Tertiles are equal groupings of a sample into thirds. Please change the description of the categories 10%, 10-90, >90. Line 307 this sentence is unclear. What is unchanged? what are the modifiable factors? Line 310 what is the ecosystem? This sentence is not needed and repetitive with the preceding and following sentences. Suggest to cut. Line 326 early screening MAY prevent SOME poor outcomes.
--

VERSION 1 – AUTHOR RESPONSE

Reviewer Report:

Reviewer: 1 - Comments to the Author:

This cohort paper describes the measurement of exposures and outcomes in a longitudinal birth cohort study among a difficult to reach population. The undertaking of this study is commended, and the care taken to report the measurements, prevalence of exposures and outcomes, and implications is appreciated. There are a few minor revisions that I think will improve the MS.

Response: We thank the reviewer for their detailed review of the manuscript. We appreciate all the attention to details and the suggestions that we strongly believe have improved our manuscript. We have responded point-wise to all of the reviewers' comments below, and mentioned the updated pages and lines as per the Revised manuscript with Tracked changes.

1. Strengths and limitations: Overall this section could be revised to be more streamlined. There are many redundant sentences. The vulnerability of the population and the implications of the study are repeated in almost every paragraph. This entire section could be substantially shortened.

Response: We thank the reviewer for the suggestions. We have now shortened the entire section limiting to the strengths and limitations of the study. The edits are made on pages 21 – 23.

2. Introduction: Please provide a brief background into the interest in the secondary aims (e.g., depression).

Response: Thank you for the comments. The background has been added to page 4-5, lines 96 to 104, with relevant references.

3. Tables: It may be worthwhile to provide a footnote with abbreviations and definitions. For example, what does "diseased" include, what criteria was used for GDM diagnosis? Why are there CIs for hypertension, GDM and anemia?

Response: Many thanks for the important suggestion. The suggested changes regarding the tables have been made throughout.

Confidence intervals for incident/prevalent cases provide a likely range of values of the population mean. Adding these values could help in a better understanding of the population mean distribution, hence have been included.

4. Line 254 N is reported as a %.

Response: Thank you for pointing out the typographical error. The change has now been made.

The sentence is now edited on page 13, line 255, to – “More than 90% of women were homemakers (90.0%; n=4446), while most of their spouses were unskilled workers (46.7%; n=2237%).”

5. Line 271 Tertiles are equal groupings of a sample into thirds. Please change the description of the categories 10%, 10-90, >90.

Response: Many thanks for this. We have made the suggested change on page 16, line 274.

6. Line 307 this sentence is unclear. What is unchanged? what are the modifiable factors?

Response: Thank you for pointing it out. We have now changed the sentence on page 21-22, line 311-312, to include the modifiable risk factors of obesity.

The sentence now reads, “If unchanged modifiable factors of obesity such as diet, exercise, and GDM will continue the vicious cycle of afflicting subsequent generations.”

7. Line 310 what is the ecosystem? This sentence is not needed and repetitive with the preceding and following sentences. Suggest to cut.

Response: Agreed. Many thanks for the suggestion, the same has been deleted.

8. Line 326 early screening MAY prevent SOME poor outcomes.

Response: Thank you for the suggestion. The sentence has been changed on page 22, line 332-333, and reads as follows:

“Early screening and timely management of GDM may prevent some poor outcomes for mother and child.”

VERSION 2 – REVIEW

REVIEWER	Francis, Ellen University of Colorado - Anschutz Medical Campus
REVIEW RETURNED	29-Aug-2022
GENERAL COMMENTS	The requested revisions were addressed. I will defer to the editor, but it is my opinion that the writing in the discussion could be improved to be more scientific or in a professional style.